

# Testing of microsatellite multiplexes for individual identification of Cape Parrots (*Poicephalus robustus*): paternity testing and monitoring trade

Willem G. Coetzer, Colleen T. Downs, Mike R. Perrin and Sandi Willows-Munro

School of Life Science, University of KwaZulu-Natal, Pietermaritzburg, South Africa

## ABSTRACT

**Background:** Illegal trade in rare wildlife species is a major threat to many parrot species around the world. Wildlife forensics plays an important role in the preservation of endangered or threatened wildlife species. Identification of illegally harvested or traded animals through DNA techniques is one of the many methods used during forensic investigations. Natural populations of the South African endemic Cape Parrot (*Poicephalus robustus*) are negatively affected by the removal of eggs and chicks for the pet trade.

**Methods:** In this study, 16 microsatellite markers specifically designed for the South African endemic Cape Parrot (*P. robustus*) are assessed for their utility in forensic casework. Using these 16 loci, the genetic diversity of a subset of the captive Cape Parrot population was also assessed and compared to three wild Cape Parrot populations.

**Results:** It was determined that the full 16 locus panel has sufficient discriminatory power to be used in parentage analyses and can be used to determine if a bird has been bred in captivity and so can be legally traded or if it has been illegally removed from the wild. In cases where birds have been removed from the wild, this study suggests that a reduced 12 locus microsatellite panel has sufficient power to assign confiscated birds to geographic population of origin.

**Discussion:** The level of genetic diversity observed within the captive Cape Parrot population was similar to that observed in the wild populations, which suggests that the captive population is not suffering from decreased levels of genetic diversity. The captive Cape Parrots did however have double the number of private alleles compared to that observed in the most genetically diverse wild population. This is probably due to the presence of rare alleles present in the founder population, which has not been lost due to genetic drift, as many of the individuals tested in this study are F1–F3 wild descendants. The results from this study provide a suit of markers that can be used to aid conservation and law enforcement authorities to better control legal and illegal trade of this South African endemic.

Corresponding author
Sandi Willows-Munro,
willows-munro@ukzn.ac.za

## INTRODUCTION

The illegal wildlife trade includes the buying and selling of any wildlife product that has been captured alive, poached, and used as food, medicine, pets, and trophies (*TRAFFIC, 2008*). The illegal trade in wildlife has a negative impact on wildlife and conservation programs worldwide (*Alacs et al., 2010*). The exact value of the illegal wildlife trade is unknown, but current estimates suggest that illegal transactions involving wildlife and their products is a multibillion US dollar enterprise (*Broad, Mulliken & Roe, 2002*; *Interpol, 2014*). This is particularly true for rare bird species, which are highly sought after (*Cooney & Jepson, 2006*; *White et al., 2012*). Parrots (order Psittaciformes) are extremely popular as pets and have the highest reported trade figures among all traded avian orders (*Bush, Baker & Macdonald, 2014*). Of particular concern are the rare and enigmatic species as half of the world's threatened or near-threatened parrot species are impacted by illegal trade (*Pain et al., 2006*). African parrot species are increasingly becoming targets for trade. For example, in China a quarter of all imported parrots originated from South Africa (*Li & Jiang, 2014*). To date the Convention on International Trade in Endangered Species of Wild Fauna and Flora (CITES) has classified South Africa as a major importer and exporter of legally and illegally obtained birds (*Warchol, 2004*) and is regarded as the hub of both legal and illegal wildlife trade in the region (*Wynberg, 2002*).

Captive breeding of exotic birds is a plausible alternative to sourcing wild animals, and it has been shown to be a viable practice (*Pires, 2012*). Breeding of wildlife in captivity is, however, not always an alternative to wild harvesting, as there will always be a demand for new breeding stock from the wild (*Nogueira & Nogueira-Filho, 2011*; *Bush, Baker & Macdonald, 2014*). The creation of self-sustaining captive populations, which resembles the wild genetic lineages as closely as possible, should be one of the main aims of captive-breeding programs if reintroductions are proposed (*Frankham, 2008*; *Robert, 2009*). Regular assessments of the genetic fitness of captive populations are therefore important to ensure healthy populations exist for possible reintroductions. The legitimacy of some "captive-bred" animals are also a concern, as it has been suggested that some breeding facilities produce more "captive-bred" animals than is plausible (*Lyons & Natusch, 2011*; *White et al., 2012*). It is therefore imperative to monitor the legal trade of alleged captive bred birds to identify possible illegal activities.

Molecular forensic methods are widely used to identify suspected illegally obtained wildlife or wildlife products (*Comstock, Ostrander & Wasser, 2003*; *Gupta, Verma & Singh, 2005*; *Lorenzini, 2005*; *Dawnay et al., 2009*; *Lorenzini et al., 2011*; *Coghlan et al., 2012*; *White et al., 2012*; *Mondol et al., 2014*; *Gonçalves et al., 2015*; *Presti et al., 2015*). One of the most useful molecular forensic tools is genetic fingerprinting using microsatellite markers. These markers have been used to identify legally, and illegally, traded birds when a sufficient reference database is available (*White et al., 2012*; *Presti et al., 2015*). It is necessary to consider the genetic sub-structuring within a species if the reintroductions of confiscated animals are considered, as the subpopulations could have acquired habitat specific fitness (e.g., pathogen resistance; *Boyce et al., 2011*). It is important, from a conservation viewpoint, to preserve genetically distinct or evolutionary significant

populations (*Johnson, 2000*). The use of microsatellite data to assign confiscated wildlife to their area of origin is a well-known technique used in wildlife forensic and conservation sciences (*Manel, Berthier & Luikart, 2002*; *White et al., 2012*; *Mondol et al., 2014*; *Presti et al., 2015*). For example, *Presti et al. (2015)* were able to assign 24 confiscated hyacinth macaw chicks to their populations of origin based on Bayesian clustering analysis using 10 microsatellite loci and *White et al. (2012)* were able to identify the kinship and area of origin of a white-tailed black cockatoo using 20 microsatellite loci and kinship analyses.

Several factors need to be considered when selecting a microsatellite panel for forensic studies. The quality of the data obtained from a set of markers should be assessed by considering the occurrence of genotyping errors such as null alleles and missing data, which can lead to biased estimations of genetic diversity and false parentage assignments (*Dakin & Avise, 2004*). Additionally, the level of informativeness of each marker should be assessed, focusing on the level of variation and the discriminatory power of each locus (*Rosenberg et al., 2003*). For more information on molecular methods in wildlife forensics and microsatellite null alleles, refer to the articles of *Alacs et al. (2010)* and *Dakin & Avise (2004)*.

Numerous well-established methods are available to assess the informativeness of genetic markers namely, the polymorphic information content (PIC) estimate (*Botstein et al., 1980*), the probability of identity ($P_{ID}$; *Taberlet & Luikart, 1999*; *Waits, Luikart & Taberlet, 2001*), and the probability of exclusion ($P_E$; *Fung, Chung & Wong, 2002*). The $P_{ID}$ and $P_E$ estimates are well-established methods for assessing the ability of molecular markers to distinguish between individuals (*Taberlet & Luikart, 1999*; *Fung, Chung & Wong, 2002*).

The South African endemic Cape Parrot (*Poicephalus robustus*) is a locally endangered parrot species found in the South African mistbelt forests (*Wirminghaus, 1997*; *Taylor, 2014*), with fewer than 1600 individuals left in the wild (*Downs, Pfeiffer & Hart, 2014*). It has been suggested that the Cape Parrot is under tremendous pressure, not only due to habitat fragmentation, but also due to the illegal harvesting of wild birds and eggs for the pet trade (*Wirminghaus et al., 1999*; *Martin et al., 2014*). The Cape Parrot is currently still observed as a subspecies of *P. robustus* and it was therefore not classified by the International Union for Conservation of Nature (IUCN) or CITES as endangered or threatened. Recent genetic work has, however, shown that the Cape Parrot should be elevated to species status (*Coetzer et al., 2015*), separate from the more widely distributed gray-headed parrot (*Poicephalus fuscicollis, P. f. suahelicus*).

Cape Parrots have been successfully bred in captivity for a number of years, although it is a difficult practice with low-breeding success among wild-caught breeding pairs (*Wirminghaus et al., 1999*). Captive-breeding facilities provide the pet trade with legally obtained animals, and may also serve as source populations if future reintroductions to natural habitats are needed (*Storfer, 1999*; *Williams & Hoffman, 2009*; *Pires, 2012*). The occurrence and accumulation of deleterious mutations, as well as the effects of genetic adaptation to captivity (*Williams & Hoffman, 2009*), are major issues observed in captive populations. It was recently observed that the current Cape Parrot population shows signs of genetic sub-structuring, with three genetic clusters which are geographically correlated along the Cape Parrot distribution range (*Coetzer, 2015*). It is therefore

important to maintain captive populations, which are genetically similar to these three genetic lineages if future reintroductions are needed. Proper studbook keeping and managing of the captive populations are therefore essential for maintaining healthy captive bred wildlife populations (*Ferrie et al., 2013*). The regional Cape Parrot studbook currently holds records of 341 Cape Parrots, 216 extant (*Wilkinson, 2015*). The studbook is, however, currently lacking many records due to many breeders showing reluctance in sharing information with regards to their Cape Parrot stocks (Wilkinson, 2015, personal communication).

In this study, three main aims are addressed. First, an assessment of 16 microsatellite markers previously designed specifically for Cape Parrots (*Pillay et al., 2010*) was conducted to determine their utility in forensic analyses. A subset of these 16 loci were previously used in a higher-level taxonomic analysis of *Poicephalus* parrots (*Coetzer et al., 2015*) and all 16 loci were used in a phylogeographic assessment of the Cape Parrot (*Coetzer, 2015*). Second, the utility of these 16 loci for use in assigning confiscated wild-caught birds to their area of origin was tested through a Bayesian assignment method. The approach outlined in this study will assist law enforcement and conservation authorities with the return of illegally harvested Cape Parrots to the wild. It is known that the genetic variation of populations in captivity can change markedly from the wild populations (*Hindar, Ryman & Utter, 1991*; *Lynch & O'Hely, 2001*), which can have serious implications when reintroductions are considered. It is therefore vital to assess the genetic variation and structure of the captive Cape Parrot population. Third, the genetic differentiation between the three wild Cape Parrot populations identified by *Coetzer (2015)* and the captive population was assessed using 16 microsatellite loci. These results will aid in the management of the captive population and to ensure that the captive population can be self-sustaining with minimal or no supplementation from the wild.

# MATERIALS AND METHODS

## Ethics

Ethical clearance for this study was received from the University of KwaZulu-Natal Animal Ethics sub-committee (Ref numbers: 074/13/Animal, 017/14/Animal, and 042/15/Animal). All sampling procedures followed the criteria laid out by this committee.

## Sampling and DNA extraction

Samples were collected from 76 captive Cape Parrots (Table S1). This includes samples taken from one international and five South African breeding facilities. The captive specimens included in this study comprise 22.3% of the Cape Parrot regional studbook (*Wilkinson, 2015*). The majority of these samples were sourced from one breeding facility, which holds the largest captive Cape Parrot-breeding populations in the world. Five of the specimens included in this study were wild-caught birds that were recently introduced into the captive breeding program. These five birds originated from the KwaZulu-Natal (KZN) Province. To test the utility of the molecular markers, captive birds with known pedigree were included. Both parents of 31 specimens were sampled, with only the sire sampled for seven of the captive-bred birds (Table S1).
Whole blood was collected from 45 captive Cape Parrots using Whatman™ FTA™ Elute cards (Buckinghamshire, UK) and was stored at room temperature in a dark cool storage area. Seventeen samples were whole blood stored in absolute ethanol at −20 °C. Samples were also collected from deceased birds provided by two breeding facilities ($n = 14$). Biopsies of 5 mm × 5 mm were collected from each carcass and stored in absolute ethanol at −20 °C.

DNA extraction from the Whatman™ FTA™ Elute cards (Buckinghamshire, UK) was performed following the manufacturer's protocols. The DNA extraction from the tissue and whole blood stored in absolute ethanol was performed with the NucleoSpin® Tissue kit (Macherey-Nagel, Düren, Germany), following the manufacturer's protocols. All DNA extracts were stored at −20 °C. DNA quantity was established via NanoDrop® ND-1000 Spectrophotometer (Thermo Fisher Scientific, DE, USA) analysis.

To assess any genetic differences between captive-bred and wild Cape Parrots, 85 genotypes from wild Cape Parrot populations were taken from *Coetzer (2015)*. This study assessed the phylogeographic relationships between wild Cape Parrot populations. Strong genetic sub-structuring was observed, with three distinct genetic clusters linked to three geographical regions within the Cape Parrot distribution range (*Coetzer, 2015*). The wild data set consisted of 52 genotypes from the south genetic cluster (Eastern Cape region), 19 from the central cluster (KZN region), and five genotypes from the north cluster (Limpopo region). Details on these specimens are provided in Table S2.

## Microsatellite amplification

The 16 microsatellite loci selected for this study were specifically developed for use in Cape Parrots (*Pillay et al., 2010*). The markers were divided into six multiplex sets (multiplex 1: *Prob06*, *Prob15*, and *Prob26*; multiplex 2: *Prob30* and *Prob36*; multiplex 3: *Prob18*, *Prob25*, and *Prob31*; multiplex 4: *Prob01*, *Prob09*, and *Prob17*; multiplex 5: *Prob23* and *Prob28*; multiplex 6: *Prob29*, *Prob34*, and *Prob35*). In-silico testing of all multiplexes was done prior to PCR amplification to test for the presence of primer dimers and hairpin structures using the program AutoDimer v1 (*Vallone & Butler, 2004*). The selected multiplex combinations did not show any signs of primer dimers or hairpin structures among the primer pairs. The forward primer in each microsatellite pair was fluorescently labeled on the 5′ end. The KAPA2G Fast Multiplex mix (KAPA Biosystems, Wilmington, MA, USA) was used for all amplifications, with each PCR reaction mixture consisting of: ~2–60 ng template DNA, 5 μl KAPA2G Fast Multiplex mix (KAPA Biosystems, Wilmington, MA, USA), 0.2 μM of each primer and dH$_2$O to give a final reaction volume of 10 μl. The annealing temperature prescribed by the KAPA2G Fast Multiplex kit's manufacturers was initially tested and provided positive results for all multiplex sets. The PCR cycling conditions consisted of an initial denaturation step for 3 min at 94 °C followed by 30 cycles at 94 °C for 30 s, 60 °C for 30 s, 72 °C for 30 s, and a final extension step for 5 min at 72 °C. The whole PCR setup, excluding the DNA addition step, was performed in a DNA-free area.

The amplified PCR products were sent to the Central Analytical Facility at Stellenbosch University, South Africa, for fragment analysis, using a standard ROX 500 internal size standard. The software package GeneMarker® v2.4.0 (Soft Genetics, State College, PA, USA)

was used for all genotype scoring. We reamplified 20% of the data set to check for genotype consistency.

## Data analysis

### Evaluating best set of microsatellite loci

The expectation maximization (EM) algorithm for detection of null allele frequencies was used as implemented in the software program FreeNA (*Chapuis & Estoup, 2007*). To assess the informativeness of each locus, the PIC and the allelic richness (Ar) of each locus were calculated using Cervus (*Kalinowski, Taper & Marshall, 2007*) and FSTAT (*Goudet, 2001*), respectively. Per marker, PIC values higher than 0.6 are generally seen as highly informative (*Mateescu et al., 2005*). The rarefaction method was followed for the Ar estimation to account for differences in sample size. The $P_{ID}$ and probability of exclusion (one parent known, $P_{E2}$) were estimated in GenAlEx (*Peakall & Smouse, 2012*) to evaluate the discriminatory power of each locus. The combined $P_{ID}$ and $P_{E2}$ values were also calculated. Deviations from Hardy–Weinberg equilibrium (HWE) were estimated using Genepop v4.2 (*Rousset, 2008*).

Each locus was ranked according to their null allele, PIC, Ar, and $P_{ID}$ and $P_{E2}$ estimates (Table 1). A score of 1 (excellent) to 16 (poor) was given to each locus for each of these five estimates, with a minimum of 5 (highly informative) to a maximum of 80 (highly uninformative). Eight microsatellite panels were then assembled by selecting the highest-ranking loci for each panel, containing 9–16 loci each (Table S3). Each of these eight panels was tested in the subsequent parentage and assignment analyses.

## Parentage testing of captive population

The eight selected microsatellite panels were evaluated by testing the accuracy of each panel for parentage assignments using captive specimens with both known and unknown pedigrees. For this analysis, the full-pedigree maximum likelihood method implemented in COLONY v2.0.4.6 (*Jones & Wang, 2010*) was used. This program compensates for genotyping errors and null alleles (*Wang, 2004*) and has been used previously to identify parentage in captive (*Ferrie et al., 2013*; *Loughnan et al., 2015*) and wild vertebrate populations (*Masello et al., 2002*; *Riehl, 2012*; *Bergner, Jamieson & Robertson, 2014*). The offspring data set consisted of 38 individuals. For seven of these, only the paternal parent was known. A monogamous mating system with no inbreeding was selected, using the full-likelihood method. A medium run length with no prior sibship selected. The marker type and null allele frequencies for each locus were uploaded with an error rate of 0.02 as suggested by *Wang (2004)*. We uploaded the genotypes of 38 offspring, 30 paternal candidates, and 21 maternal candidates, with the probability of the sire or dam included in the data set at 0.75 and no paternal or maternal exclusion information.

## Power of microsatellite panel to detect origin of illegally traded birds

A partial Bayesian exclusion approach (*Rannala & Mountain, 1997*) as implemented in GenClass2 (*Piry et al., 2004*) was used to further assess the eight microsatellite panels.
**Table 1 Primer details and genetic diversity estimates per locus as calculated from the captive *Poicephalus robustus* data set used in the current study.** The standard error for the average number of alleles is provided in parentheses. The values in superscript indicates the locus' rank for the specific measure (1 = excellent and 16 = poor). It should be noted that *Prob15* is reportedly Z-linked (Pillay et al., 2010) which could influence the null allele frequency.

| Locus | Primer sequence (5′–3′) | Average number of alleles ($N_A$) | Allelic richness (Ar) | Probability of identity ($P_{ID}$) | Probability of exclusion (one parent known; $P_{E2}$) | Polymorphic information content (PIC) | Null allele frequency as % (Na): | Inbreeding coefficient ($F_{IS}$) | Hardy-Weinberg deviation (p-value) | Locus rank |
|---|---|---|---|---|---|---|---|---|---|---|
| *Prob17* | F: TGAACATGACTTATTTGTCTAGTCATACCTAATCC<br>R: TTCCAAGGAGTAATATACAGATAATTGCTTCTACA | 17 (3.559) | 22[1] | 0.018[1] | 0.658[1] | 0.888[1] | 0.022[3] | 0.017 | 0.045 | 1 |
| *Prob31* | F: GCTGCAGTACAGGCAGTCTTTG<br>R: CCCATGGCAGAAATTACAGTGA | 5.25 (1.109) | 6.997[5] | 0.08[2] | 0.404[2] | 0.746[3] | 0.00[1] | −0.058 | 0.096 | 2 |
| *Prob26* | F: GATCCCCAAAACAGATGAGTCT<br>R: GTTTCTTGATTCAGATTGGAGGCTGATG | 7.25 (1.436) | 9.877[3] | 0.088[4] | 0.370[4] | 0.723[4] | 0.00[1] | −0.109 | 0.188 | 3 |
| *Prob30* | F: ACACTGAACCATGTCACACAAG<br>R: GATCAGAAGGCTGCTTTGC | 6 (0.707) | 5.997[8] | 0.081[3] | 0.397[3] | 0.751[2] | 0.037[4] | 0.044 | 0.0001* | 4 |
| *Prob23* | F: CACCAGTCATGACAGATAAT<br>R: AGTATAAATTCAGCCTAGTTATGT | 5 (1.08) | 5.997[7] | 0.106[5] | 0.341[5] | 0.707[5] | 0.01[2] | −0.091 | 0.034 | 5 |
| *Prob25* | F: GATCCAGTGTGAAGCTAAAACAAGG<br>R: GTTTCTTAAGGTAGATGTGGGAGTGTAG | 4.75 (0.629) | 5.946[9] | 0.113[6] | 0.330[6] | 0.691[7] | 0.00[1] | −0.028 | 0.695 | 6 |
| *Prob18* | F: GATCATTGAGAACTATTTGGAAG<br>R: GTTTCTTATCAGTTGAACGCGAGAA | 4.25 (0.479) | 5[10] | 0.112[7] | 0.327[7] | 0.694[6] | 0.00[1] | 0.035 | 0.198 | 7 |
| *Prob06* | F: TCCAACCCACCTGAATTATCCAT<br>R: GTTTCTTAGCTCCAATTCCGGGCTCT | 6 (1.414) | 7.957[4] | 0.197[9] | 0.213[9] | 0.566[9] | 0.00[1] | −0.022 | 0.606 | 8 |
| *Prob09* | F: GAACGTTTGTAGGGATAGTCCAC<br>R: GTTTCTTACCGTGTCCACCCCTTATTCG | 7.25 (1.493) | 10.833[2] | 0.199[10] | 0.198[10] | 0.56[10] | 0.06[6] | 0.146 | 0.003* | 9 |
| *Prob15* | F: GTGTCCCAGCCAGACCAAT<br>R: TCAGGTGTCCGTGTCCTGCTTCC | 5.5 (1.323) | 6[6] | 0.135[8] | 0.303[8] | 0.656[8] | 0.186[9] | 0.439 | 0* | 10 |
| *Prob01* | F: TGCTCCCCATTCTACAGGTC<br>R: TGTTTCCATAATTTGGCTTGC | 3 (0.408) | 3.999[14] | 0.207[11] | 0.186[11] | 0.559[11] | 0.058[5] | 0.129 | 0.016 | 11 |
| *Prob29* | F: CAACACTGTGTATGCCCATGC<br>R: GTTTCTTGTTTGGACCCAGCAATCACC | 3.75 (0.629) | 4[13] | 0.338[13] | 0.108[13] | 0.415[13] | 0.00[1] | −0.1 | 0.134 | 12 |
| *Prob34* | F: GGTGCTGGAAGGTGGCTTCT<br>R: GCTTTGGCTGGTGGTCCATT | 4 (0.408) | 4.999[11] | 0.363[14] | 0.095[14] | 0.392[14] | 0.00[1] | −0.055 | 0.004 | 13 |
| *Prob28* | F: GATCAAGGTATCATTAATAAGC<br>R: GAGCTCTCATTGTATGTCAA | 3 (0.707) | 4.957[12] | 0.28[12] | 0.167[12] | 0.475[12] | 0.081[7] | 0.14 | 0.001* | 14 |
| *Prob35* | F: ATTTGCTGTATTGTGGGTAGG<br>R: GATCAGCTCTTCACAGGAAT | 2.5 (0.5) | 3.995[15] | 0.557[15] | 0.034[15] | 0.246[15] | 0.00[1] | 0.055 | 0.067 | 15 |
| *Prob36* | F: GATCAAAAGCTATCTGACTGGACA<br>R: GTTTCTTCCATATTCTCATTTGCTTTC | 1.75 (0.25) | 2[16] | 0.59[16] | 0.032[16] | 0.221[16] | 0.125[8] | 0.471 | 0.001* | 16 |
| Mean | | 6.813 (1.089) | 6.910 (1.149) | – | – | 0.581 (0.047) | 3.62 (1.37) | 0.047 | – | |

Note:
* p-value < 0.003.

This method estimates the likelihood that a test sample belongs to one of the reference populations provided for analysis and calculates a sample exclusion probability for each of the reference populations (*Ogden & Linacre, 2015*). The use of assignment methods to identify the population or area of origin of confiscated wildlife is a well-known tool in wildlife forensics (*Alacs et al., 2010*; *Ogden & Linacre, 2015*). The partial Bayesian assignment analysis implemented in GeneClass2 is a well-established method for assignment of specimens to their population of origin (*Primmer, Koskinen & Piironen, 2000*; *Grobler et al., 2005*; *Pruett et al., 2010*). All wild born individuals were excluded from the captive population reference data set. To simulate an assignment study, six captive-bred and six wild-caught individuals were selected at random for the "samples to be assigned" data set. The captive population from the current study and the three wild populations from *Coetzer (2015)* were used as reference populations. The 12 individuals selected for the "samples to be assigned" data set were excluded from these data sets. The Bayesian method from *Rannala & Mountain (1997)* was followed, with probability computation using Monte-Carlo resampling and the simulation algorithm from *Paetkau et al. (2004)*. The number of simulated individuals was set at 100,000, with the Type I error set at 0.01 and the assignment threshold at 0.05. These parameters were used for each of the eight microsatellite panels.

### Captive vs wild Cape Parrots

The genetic diversity of the captive bred sample group was compared to the three wild Cape Parrot populations identified in the recent phylogeographic study (*Coetzer, 2015*). Values compared included the average number of alleles ($N_A$), number of private alleles ($P_A$), observed heterozygosity ($H_O$) and unbiased expected heterozygosity ($uH_E$) estimated in GenAlEx, inbreeding coefficient ($F_{IS}$) using Genepop v4.2 (*Rousset, 2008*) and Ar using rarefaction in FSTAT (*Goudet, 2001*). A Wilcoxon signed-ranked test was performed to assess the level of genetic differences between the captive population and the three wild populations using the per locus estimates for each of the $N_A$, $H_O$, $uH_E$, $F_{IS}$, and Ar estimates. Pairwise $F_{ST}$ values and analysis of molecular variance (AMOVA) were estimated to assess if the captive population constitutes a separate genetic unit using GenAlEx. Bonferroni correction was implemented to all *p*-values to correct for problems with multiple testing (*Rice, 1989*). In this analysis, the five wild born individuals were removed from the captive data set.

## RESULTS

### Marker analysis

For this study, 76 captive Cape Parrots were successfully genotyped across 16 microsatellite loci. Loci were amplified across a range of DNA template concentrations, with low template concentration (2–5 ng/μl) successfully amplifying with minimal signs of allelic dropout (<3% over all loci). The replicate genotypes did not show any signs of discrepancies. Co-amplification of each multiplex was also highly successful, with the most amplification failures (4 of 76) observed for the locus with the largest bp size (*Prob17*). The data set contained less than 1 missing data, with a mean null allele

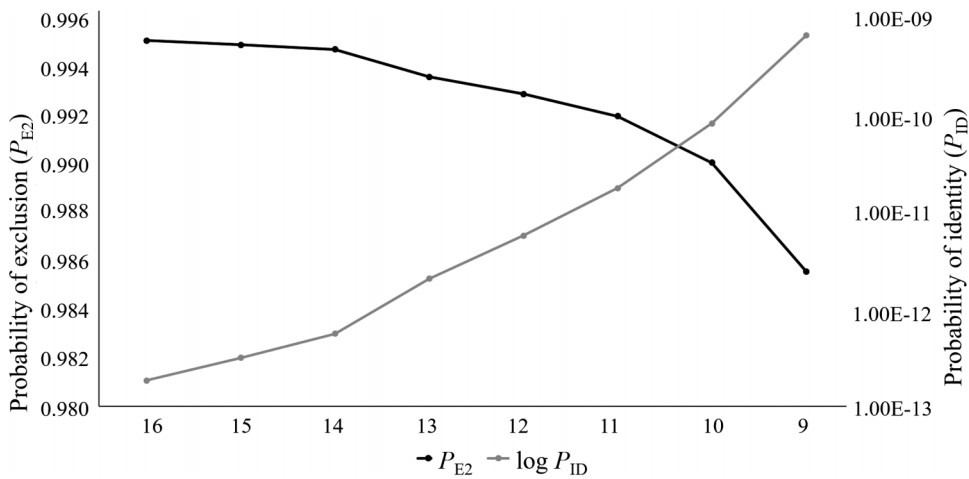

**Figure 1** **The log values of the probability of identity (log $P_{ID}$) and probability of exclusion (one parent known, $P_{E2}$) estimates for the eight microsatellite panels tested on the captive *Poicephalus robustus* data set in the current study.** It can be observed that the full 16 locus panel has the most optimum $P_{ID}$ and $P_{E2}$ values compared to the other seven panels tested in this study.

frequency over all loci and samples of 0.039. The per locus null allele frequencies ranged from 0 to 0.186 (Table 1). Only two loci showed null allele frequencies above 0.1 (*Prob15*, Na = 0.186; *Prob36*, Na = 0.125). The mean number of alleles per locus varied greatly among loci, ranging from 1.75 (*Prob36*) to 17 (*Prob17*) alleles. A large difference in Ar values was observed across the loci, with values ranging from 2 (*Prob36*) to 22 (*Prob17*). Seven loci showed high levels of heterozygosity (Table 1), with negative $F_{IS}$ values. Only two loci showed signs of heterozygote deficiency (*Prob09*, $F_{IS}$ = 0.439; *Prob36*, $F_{IS}$ = 0.471). Fourteen of the 16 loci were moderately to highly informative, with values ranging from 0.415 (*Prob29*) to 0.888 (*Prob17*). Only two loci (*Prob35* and *Prob36*) were identified as uninformative (PIC < 0.3; Table 1). The $P_{ID}$ values ranged from 0.019 (*Prob17*) to 0.591 (*Prob36*). A combined $P_{ID}$ over all 16 loci was calculated as 1.831E−13 following the product rule. The $P_{E2}$ ranged from 0.658 (*Prob17*) to 0.032 (*Prob36*), with the combined $P_{E2}$ at 0.995. It was observed that the $P_{ID}$ and $P_{E2}$ values improved as the number of loci analyzed increase (Fig. 1). Five of the 16 loci significantly deviated from HWE (*Prob09, Prob15, Prob28, Prob30,* and *Prob36*), following Bonferroni correction ($p < 0.003$). Of the 120 per locus comparisons made during the linkage disequilibrium (LD) analysis, more than half of the locus pairs showed signs of LD (52.5%).

## Parentage analyses

All eight microsatellite panels showed very low combined $P_{ID}$ values, with moderate to high informativeness levels (PIC range: 0.581–0.703; Table S4). The $P_{ID}$ values for the eight panels ranged from 1.8E−13 for the 16 locus panel to 5.7E−10 for the nine locus panel (Table S3). These values suggest that 1 in 5.5E+12 (16 loci) to 1.8E+9 (nine loci) randomly chosen individuals will share the same genotype. The assessment from this parameter alone suggests that any of these panels could be suitable for forensic use, as the total number of wild Cape Parrots does not exceed 1,600 individuals. The ability of these eight panels to successfully identify known parents, however, differed. The seven

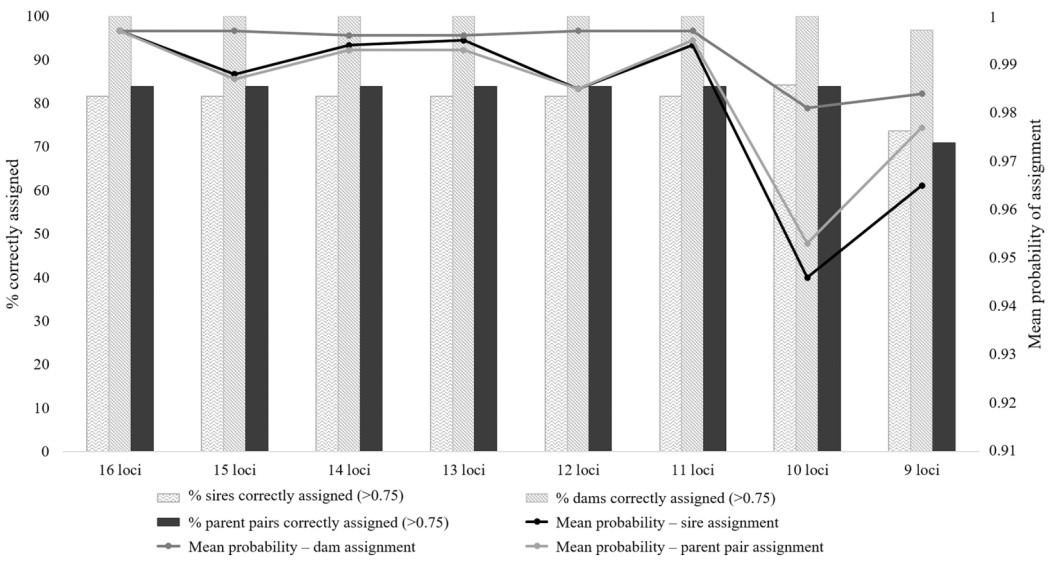

**Figure 2 The parent pair and individual parentage assignment success of the eight microsatellite panels tested for use in *Poicephalus robustus*.** The bars correspond to the percentage of known parents correctly assigned to each offspring, with high probability; the lines are representative of the probability values for each assignment type.

larger panels were generally equally successful in identifying parent pairs and individual parents, with only slight differences in the mean probability values and a slightly higher sire identification success rate for the 10 locus panel (Fig. 2; Table S4). The nine-locus panel was less successful in correctly identifying parent pairs, with only 71% of parent pairs correctly identified with high probability ($p > 0.75$). The nine-locus panel also showed a lower success rate at identifying the correct sires and dams, with 73.7% of sires and 96.8% of dams correctly identified with high probability ($p > 0.75$). All known dams were correctly identified using the seven larger panels. Although the seven larger panels had similar assignment success rates (parent pair assignment success = 83.9%), the full 16 locus panel had overall higher mean probability rates making this panel most suited for use in future parentage analyses.

The majority of the test individuals were assigned to the correct population of origin following the partial Bayesian exclusion analyses using the eight microsatellite panels (83.33%–66.67%). The highest assignment success was achieved with the six larger microsatellite panels (16 loci to 11 loci), with 83.33% of the specimens correctly assigned (Fig. 3; Table S5). The 12 locus panel had the best average assignment probability value out of the eight tested panels (average assignment probability = 0.565, SE = 0.087), with 5 out of the 10 individuals correctly assigned with assignment probabilities above 0.6. The remaining five individuals were assigned to the correct populations with assignment probabilities lower than 0.6 (assignment probability = 0.170–0.591; Table S5). The two individuals (*FH12* and *FH32*) that were incorrectly assigned, were sampled from the Eastern Cape but assigned to the captive (*FH12*) and KZN (*FH32*) populations. The assignment probabilities of these individuals did, however, differ only slightly between the actual population of origin and the assigned population (Table S5).

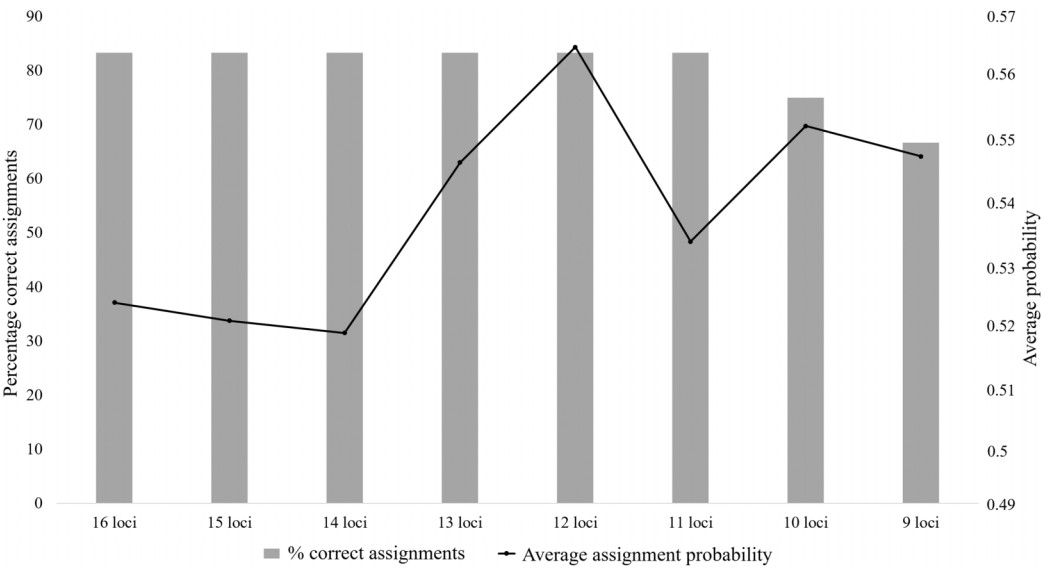

**Figure 3 The assignment success rates for the eight microsatellite panels following the partial Bayesian exclusion analyses performed to assign *Poicephalus robustus* specimens to their area of origin.** The black line represents the average assignment probability calculated from the correctly assigned specimens' probability values. The exact probability values for the assignments conducted with each of the eight panels are available in Table S5.

**Table 2 The genetic diversity estimates for each of the wild *Poicephalus robustus* populations and the captive data set, using 16 microsatellite loci.** Standard error for average number of alleles, observed heterozygosity and unbiased expected heterozygosity is provided in parentheses.

| Locality | Average number of alleles ($N_A$) | Allelic richness (Ar) | Observed heterozygosity ($H_O$) | Unbiased expected heterozygosity ($uH_E$) | Inbreeding coefficient ($F_{IS}$) | Private alleles ($P_A$) |
|---|---|---|---|---|---|---|
| South | 6.563 (1.252) | 3.791 (0.400) | 0.605 (0.055) | 0.632 (0.053) | 0.042 | 13 |
| Central | 5.313 (0.898) | 3.708 (0.386) | 0.647 (0.058) | 0.635 (0.05) | −0.02 | 5 |
| North | 2.875 (0.34) | 2.875 (0.340) | 0.6 (0.063) | 0.572 (0.052) | −0.055 | 2 |
| Captive | 6.813 (1.089) | 3.673 (0.314) | 0.591 (0.065) | 0.625 (0.047) | 0.054 | 21 |

## Genetic diversity: captive vs wild populations

The genetic diversity estimates for the captive data set, using 16 microsatellite loci, were largely similar to that observed for the wild Cape Parrot populations (Table 2). Significant differences were observed between the captive/central and captive/north per locus Na estimates ($p < 0.003$). The average number of alleles observed for the captive data set was higher than that observed in the wild population (captive, $N_A = 6.813$; southern, $N_A = 6.563$; central, $N_A = 5.313$; northern, $N_A = 2.875$; Table 2). The significant differences observed for the Na estimates could, however, be influenced by differences in sample size (captive born samples, $n = 71$; south, $n = 60$; central, $n = 20$; north, $n = 5$). The Ar estimates provide a more accurate estimation, with no significant difference observed among the captive and wild per locus Ar estimates ($p < 0.003$). The captive data set did, however, have the highest number of private alleles ($P_A = 21$), which was almost double that of the highest value observed among the wild populations (southern, $P_A = 13$). All private alleles occurred at low frequencies, with private allele frequencies for the captive data set ranging from 0.007 to 0.100 and frequencies for the southern wild

population ranging from 0.008 to 0.133. No significant differences were observed between the captive vs wild per locus observed $H_O$, unbiased expected $uH_E$ and $F_{IS}$ comparisons.

The $H_O$ for the captive data set was only slightly lower than that observed for the three wild populations (captive, $H_O = 0.591$; southern, $H_O = 0.605$; central, $H_O = 0.647$; northern, $H_O = 0.6$), with an $uH_E$ comparable to that observed for the south and central wild populations (captive, $uH_E = 0.625$; southern, $uH_E = 0.632$; central, $uH_E = 0.635$). A low positive $F_{IS}$ value was observed for the captive data set indicating only slight inbreeding ($F_{IS} = 0.054$), with low heterozygote deficiency. Low genetic differentiation was found only between the captive data set and the south population ($F_{ST} = 0.017$; $p = 0.001$). No significant genetic differentiation was observed between the captive and north populations ($F_{ST} = 0.104$; $p = 0.004$) or the captive and central populations ($F_{ST} = 0.01$; $p = 0.024$), following a Bonferroni correction ($p = 0.003$). The global $F_{ST}$ value calculated for the captive and three wild populations did not significantly differ from zero ($F_{ST} = 0.008$; $p = 0.008$). AMOVA indicated that 92% of the observed genetic variance occurred within individuals, with 5% of the genetic variance between individuals and only 3% among the populations.

## DISCUSSION

### Microsatellite evaluation

The high LD levels observed in the current study was, however, not observed during the phylogeographic analysis of Cape Parrots by *Coetzer (2015)*. This could be explained by the population history of the captive population, as admixture, rapid population expansion, bottleneck events, and genetic drift can affect LD (*Rogers, 2014*). The selection of a suitable locus combination was therefore not based on these LD estimates. The null allele frequencies and expected heterozygosity values observed for the loci from the current study is generally comparable to that reported by *Pillay et al. (2010)*. The majority of these loci were highly to moderately informative, following the current study. The Ar values from the current study only marginally deviated from the allele numbers reported by *Pillay et al. (2010)*. Locus *Prob15* was previously reported as Z-linked in Cape Parrots (*Pillay et al., 2010*). Two of the known females for the current study were, however, heterozygous at this locus. The same trend was observed by *Taylor (2011)* who found no evidence of sex linkage of *Prob15* in other *Poicephalus* species, including *P. fuscicollis fuscicollis* and *P. f. suahelicus*. The level of variation and informativeness observed in the current study is comparable to that observed in other studies. The PIC values observed in the current study fall within the same range as the values observed by *Klauke, Segelbacher & Schaefer (2013)* during a study investigating the breeding system of the endangered El Oro parakeet from southwest Ecuador. The combined $P_{ID}$ observed by *Russello et al. (2007)* for 14 loci used in the South American Monk parakeet was lower than that observed for Cape Parrots in the current study.

### Microsatellite loci for parentage analysis

The locus informativeness analyses performed on the 16 microsatellite loci allowed for the ranking of the 16 loci according to their level of informativeness. The full locus set
of 16 markers showed to be the best combination of markers for parentage analysis of the eight panels tested. This panel had the highest average assignment probabilities for parent pair, sire, and dam assignment tests (Fig. 2). This panel is highly suited for individual-level identification, with a $P_{ID}$ value suggesting that 1 in 5.5E+12 individuals will share the same genotype. All the known dams were positively identified with this panel, but only 31 of the 38 offspring's known sires were identified with high probability. It was observed that the 10 locus panel did have a better assignment rate for the known sires, with 32 of the 38 sires identified, but the average assignment probability value was much lower than that observed for the 16 locus panel (Fig. 2). Similar success rates were observed by *Labuschagne et al. (2015)* when assessing the utility of 12 microsatellite loci for parentage assignment in the African Penguin (*Spheniscus demersus*). The failure to assign a parent, or assignment of a parent with low probability, can be linked to the occurrence of amplification errors during PCR causing allelic dropout, null alleles, stuttering, or polymerase slippage (*Buckleton, Triggs & Walsh, 2005*; *Dewoody, Nason & Hipkins, 2006*; *Ferrie et al., 2013*). It is possible to compensate for such errors in the parentage analysis program COLONY by importing the expected error rates of each locus, including null allele frequencies, prior to analysis. When this was done in the current study, failed or incorrect assignments were still observed and it is advisable to not only rely on genetic data, but also make use of a complete studbook of legally registered captive bred birds, as suggested by *Ferrie et al. (2013)*. It is therefore important to compile a complete studbook of all captive bred Cape Parrots, complemented by a complete DNA database using the full 16 locus microsatellite panel described in this study. The inclusion of additional markers such as single nucleotide polymorphisms (SNPs) can improve the success rate of the parentage analysis, as demonstrated in *Labuschagne et al. (2015)*.

## Population of origin analysis

The assignment analysis performed with six of the eight microsatellite panels (16 loci– 11 loci) all had a success rate of 83.33%. The average assignment probabilities of the correct assignments did however differ, with the 12 locus panel outperforming the rest (Fig. 3). The two Eastern Cape individuals were not assigned to their population of origin, with sample *FH12* assigned to the captive population and *FH32* assigned to KZN (Table S5). The probabilities that these two samples should be assigned to the captive and KZN populations were only marginally higher than the probabilities observed for assignment to the Eastern Cape population. This could be due to the occurrence of ancestral admixture, as it is shown that the southern (Eastern Cape) populations form a source to the central (KZN) populations (*Coetzer, 2015*), and the captive-bred populations in turn is largely sourced from the KZN populations (C. T. Downs, 2015, unpublished data). These individuals could therefore have ancestral links to individuals in the central (KZN) and the captive populations (via the KZN populations).

## Captive vs wild Cape Parrots

The majority of the genetic differentiation estimates showed little to no genetic difference between the captive data set and the three wild Cape Parrot populations. Similar

AMOVA results were observed in a study focusing purely on the genetic variation among the wild Cape Parrot populations (*Coetzer, 2015*). A clear difference was, however, observed for the private allele estimate. The captive data set contained almost double the number of private alleles observed in the southern wild population, which the recent phylogeographic study suggests is the most genetically diverse wild population (*Coetzer, 2015*). In theory, the higher number of private alleles in the captive population could be due to rare alleles, which are generally not often seen in the wild, being present in the founders of the captive populations. *Gautschi et al. (2003)* observed a similar trend in a captive bearded vulture (*Gypaetus barbatus*) population, with a higher level of genetic diversity in the captive population when compared to that observed in the wild population. This was linked to founder individuals, who are still present in the breeding population, carrying rare alleles and thereby passing these alleles down to their offspring (*Gautschi et al., 2003*). It is therefore possible that the captive Cape Parrots have not been in captivity for an appropriate amount of time to lose the observed rare alleles, and that these alleles are still being passed down to the new generations. New wild birds are also regularly introduced to the captive stock, through the addition of injured or confiscated wild birds (*Wilkinson, 2015*). These introductions could then also supplement the genetic diversity of the captive population, especially if the birds originate from different regions of the Cape Parrot's natural distribution range. Many of the birds in the captive data set used in the current study are F1–F3 descendants from wild birds, and could therefore still carry these rare alleles.

The captive and wild birds are also subjected to different environmental forces, which can lead to genetic adaptation to captivity (*Frankham, 2008*). The difference in selective pressures such as a lack of predators, food availability, and pathogenic exposure could promote the selection of certain traits in captive animals, which would normally be detrimental in the wild (*Lynch & O'Hely, 2001*). It is possible for selection of certain rare, fitness-linked, loci to influence the genetic diversity of neutral loci like microsatellites, although it was observed to mostly decrease the genetic diversity of neutral loci (*Montgomery et al., 2010*). It could, therefore, also be argued that the large number of private alleles observed in the captive sample set could in some way be linked to the selection of rare alleles, due to human mediated mate selection of breeding pairs. Further analyses using fitness-linked loci, like the major histocompatibility complex (MHC) genes or toll-like receptor (TLR) genes, should be performed to better understand the effects captive breeding has on the genetic health of the Cape Parrot population.

## CONCLUSION

The assessment of the 16 microsatellite loci tested in the current study identified the full 16 locus panel as the best set of markers for use in parentage analysis. Such analyses should be performed on traded birds suspected of being illegally harvested from the wild. It is therefore important to have a database of all legally owned Cape Parrots and a complete studbook for future use. Using this set of loci, birds suspected of being illegally harvested from the wild can be traced to the region of origin through implementation of

the partial Bayesian approach in GeneClass2 for individual assignment analysis. The 12 locus microsatellite panel is most appropriate for this analysis. It is, however, recommended to increase the reference data sets, for both the wild and captive populations, thereby increasing the accuracy of the individual assignment analysis using the assignment methods implemented in GeneClass2. This recommendation is based on the low level of differentiation observed between the wild and captive populations. The use of additional highly polymorphic loci could improve these results (*Cornuet et al., 1999*). The high number of private alleles observed in the captive population highlights its distinctiveness. Reintroductions to the wild from the current captive population is not recommended until further analyses of fitness related loci are performed, as accumulation of certain rare alleles could have detrimental effects on the wild populations. It is further recommended that, for reintroduction purposes, captive populations from the three Cape Parrot populations should be kept separate to prevent unnatural admixture of the different genetic groups.

The results from this study will help conservation and law enforcement authorities to better police and identify cases of illegal trafficking in South Africa's only endemic parrot. The information obtained here also highlights the genetic distinctiveness of the captive population, and the effect these birds will have on wild populations should be considered before any future reintroductions plans are made.

## ACKNOWLEDGEMENTS

The researchers would like to thank all the private Cape Parrot breeders for their support in providing samples for this study; Mr. W. Horsfield (Amazona Birds Cc), Mr. T. Davies and Mrs. L. Davies (private breeders) and Mr. S. Wilkinson (*Montecasino* Bird Gardens, Johannesburg).

### Funding
This project was funded by University of KwaZulu-Natal, National Research Foundation of South Africa and the South African National Biodiversity Institute. The funders had no role in study design, data collection and analysis, decision to publish, or preparation of the manuscript.

### Grant Disclosures
The following grant information was disclosed by the authors:
University of KwaZulu-Natal.
National Research Foundation of South Africa.
South African National Biodiversity Institute.

### Competing Interests
The authors declare that they have no competing interests.

## Author Contributions

- Willem G. Coetzer conceived and designed the experiments, performed the experiments, analyzed the data, wrote the paper, prepared figures and/or tables, and reviewed drafts of the paper.
- Colleen T. Downs contributed reagents/materials/analysis tools, reviewed drafts of the paper.
- Mike R. Perrin contributed reagents/materials/analysis tools, reviewed drafts of the paper.
- Sandi Willows-Munro conceived and designed the experiments, contributed reagents/materials/analysis tools, and reviewed drafts of the paper.

## Animal Ethics

The following information was supplied relating to ethical approvals (i.e., approving body and any reference numbers):

Ethical clearance for this study was received from the University of KwaZulu-Natal Animal Ethics sub-committee (Ref numbers: 074/13/Animal, 017/14/Animal, and 042/15/Animal). All sampling procedures followed the criteria laid out by this committee.

## Data Deposition

The raw data has been supplied as Supplemental Dataset Files.

## Supplemental Information

Supplemental information for this article can be found online at http://dx.doi.org/10.7717/peerj.2900#supplemental-information.

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
