# Peer review of "Testing of microsatellite multiplexes for individual identification of Cape Parrots (Poicephalus robustus): paternity testing and monitoring trade"

_PeerJ, doi:10.7717/peerj.2900_

## Round 0.1 · original submission · Major Revisions

Dear authors

Your ms has been reviewed and as you see the reviewers recommend a thorough revision. Please address their comments.

Kind regards,

Michael Wink
Academic editor

Reviewer 1 ·

Basic reporting

There is some mixing in the structure of the MS. Some parts of the Results belong to the Discussion.

The quality of the figures is poor. I made suggestions to the authors regarding how to improve them.

Captions to the tables and figures need to be rewritten including more information on the studied species, etc.

Please, see my comments to the authors for further details.

Experimental design

No Comments

Validity of the findings

No Comments

Additional comments

Lines 66-69: I suggest moving these lines to the paragraph starting in current line 121 to improve the flow of the Introduction.

Line 89: I suggest removing “for example”. It is clear that the references given are an example and not an exhaustive list.

Lines 111-120: I suggest moving this paragraph to Methods if specifically needed. If not, a few references will suffice to make the point.

Lines 121-123, 126-128: this information on the taxonomic status of Cape Parrots is confusing for non-specialised readers. A much better way is used in the abstract of Coetzer et al. 2015. Please, rewrite.

Line 128-132: this is an opinion/view that not all colleagues share. Some colleagues think that evolutionary significant unit (ESU) are a better tool. I would suggest not getting into this discussion here, as it is not relevant to the objectives of the study. Besides, this issue is adequately addressed in Coetzer et al. 2015.

Line 132-134: this sentence reads unclear. Please, check. Also, I would suggest using a separate paragraph for matters related to the captive breeding of Cape Parrots.

Line 134: I would suggest using “pet trade” instead of “wildlife pet trade” to avoid confusion.

Line 150: I think the first aim should be expressed in a clearer manner. Please, check.

Line 154: I think the authors mean “were”.

Lines 161-164: I would suggest placing these sentences before stating the third aim.

Lines 205-6: loci names are already given in Table 1 and thus not needed here. I would suggest deleting them from this sentence.

Line 226: a comma is missing

Lines 305-309, 326-331, 361-363, 414-415, and 424-426: I suggest moving these sentences (information from the literature or interpretation) to the Discussion.

Figs. 1, 2 and 3: the quality of these figure is poor e.g. too many labels in the y-axe, the use of horizontal lines, the use of a frame, the use of commas for decimal places, the use of colons in the axes labels, unnecessary use of colour, the use of small fonts, the black line in Fig. 3 appear to be cut. I strongly suggest redrawing the figures. Figure captions should have an indication to the species considered, as search machines usually present them separately from the accompanying text.

Lines 446-453: I suggest deleting this paragraph as it repeats information already given.

Discussion: I would suggest presenting first your main results (without giving numbers already presented in Results e.g. lines 480-481) and then compare with previous studies. In this way the value of your work will be highlighted and placed in the context of previous work. Doing the other way round dilutes the value of your work. Additionally, I would strongly suggest reducing the extent of the Discussion. This will allow the readers to concentrate on the important results, which are diluted among many comments related to previous work in the current version of the MS.

Lines 502-516: I think this is too much details and a too long justification of GeneClass2. This software is widely thus, one sentence and 2-3 references will suffice to justify its use. Besides, the justification of its use belongs to Methods not to the Discussion.

Line 527: a co-author cannot provide a “pers. comm.”. I guess the authors mean “unpublished data”.

Reviewer 2 ·

Basic reporting

The standard of the language is clear and unambiguous with the exception of some typographical or grammatical corrections as pointed out in the annotated pdf.
The introduction and background are sufficient although the flow of some sections should be improved. The relevant literature has largely been included.
The structure is appropriate including relevant illustrations. The referencing style does not conform with the journal guidelines. The authors should avoid discussion in the results section as well as repeating results in the discussion.
This is a coherent unit of publication. Even though impact is not a consideration, the study contributes to parrot conservation broadly and to wildlife forensics in general.

Experimental design

The research aims are clearly defined. However, validating loci based on the captive population is a conceptual concern. I would suggest validating the loci using a random sample of the wild population and then applying the markers to addressing the question of levels of genetic variation in the captive population, sources of the captive population and the best set of markers (from the validation on the wild population), for future forensic use.
The sample size used should be clarified.
The statistical approaches employed are standard. The investigation is rigorous and of a high technical standard. The methods are repeatable. The study adheres to ethical considerations.
All symbols for genetic parameters should follow standard conventions, e.g. the F in FST should be in italics.

Validity of the findings

The data are robust. Even though the overall result in terms of the power of the loci will likely not change significantly based on samples from the wild population, the captive population adds biases in the data.
The authors should discuss why the sire assignments are lower than that of the dams.
The Discussion and Conclusions clearly relate to the stated aims.

Additional comments

No additional comments.

Annotated reviews are not available for download in order to protect the identity of reviewers who chose to remain anonymous.

---

## Round 0.2 · Minor Revisions

Dear authors

As you can see from the remarks of our reviewer a few remaining improvements are still required.

Greetings
M Wink

Reviewer 1 ·

Basic reporting

I read it carefully and I found that all but one of my previous suggestions have been adequately addressed. In my view, the figures need further work.

Please, see my comments to the authors for further details.

Experimental design

No comments

Validity of the findings

No comments

Additional comments

Line 81: I suggest saying “have been suggested”.

Line 83: chronological order should be used.

Line 122: a double space is present.

Line 302 and elsewhere: for percentages, only decimal place suffices.

Line 322: please, move the reference to Rice (1986) to Methods, if at all needed. References are usually not needed in the Results section.

Lines 324-328: please, move to the Discussion.

Figs.: in my view, the horizontal lines are not needed but, axes (x, y) are needed. This will improve clarity further.

Line 472: I have serious concerns regarding the quality of the work cited here. I would strongly suggest looking for alternatives.

Line 554: to “AMOVA results” corresponds “were”, please, correct

---

## Round 0.3 · accepted · Accept

Good news- your ms can now be accepted

Cheers
Michael Wink